# NeuPD—A Neural Network-Based Approach to Predict Antineoplastic Drug Response

**DOI:** 10.3390/diagnostics13122043

**Published:** 2023-06-13

**Authors:** Muhammad Shahzad, Muhammad Atif Tahir, Musaed Alhussein, Ansharah Mobin, Rauf Ahmed Shams Malick, Muhammad Shahid Anwar

**Affiliations:** 1FAST School of Computing, National University of Computer and Emerging Sciences (NUCES-FAST), Karachi 75030, Pakistan; mshahzad@nu.edu.pk (M.S.); atif.tahir@nu.edu.pk (M.A.T.); k190958@nu.edu.pk (A.M.); rauf.malick@nu.edu.pk (R.A.S.M.); 2Department of Computer Engineering, College of Computer and Information Sciences, King Saud University, P.O. Box 51178, Riyadh 11543, Saudi Arabia; musaed@ksu.edu.sa; 3Department of AI and Software, Gachon University, Seongnam-si 13120, Republic of Korea

**Keywords:** NeuPD, cell lines, gene expression, machine learning, neural networks, drug response prediction, XGBoost

## Abstract

With the beginning of the high-throughput screening, in silico-based drug response analysis has opened lots of research avenues in the field of personalized medicine. For a decade, many different predicting techniques have been recommended for the antineoplastic (anti-cancer) drug response, but still, there is a need for improvements in drug sensitivity prediction. The intent of this research study is to propose a framework, namely **NeuPD,** to validate the potential anti-cancer drugs against a panel of cancer cell lines in publicly available datasets. The datasets used in this work are Genomics of Drug Sensitivity in Cancer (GDSC) and Cancer Cell Line Encyclopedia (CCLE). As not all drugs are effective on cancer cell lines, we have worked on 10 essential drugs from the GDSC dataset that have achieved the best modeling results in previous studies. We also extracted 1610 essential oncogene expressions from 983 cell lines from the same dataset. Whereas, from the CCLE dataset, 16,383 gene expressions from 1037 cell lines and 24 drugs have been used in our experiments. For dimensionality reduction, Pearson correlation is applied to best fit the model. We integrate the genomic features of cell lines and drugs’ fingerprints to fit the neural network model. For evaluation of the proposed **NeuPD** framework, we have used repeated K-fold cross-validation with 5 times repeats where K = 10 to demonstrate the performance in terms of root mean square error (RMSE) and coefficient determination (R^2^). The results obtained on the GDSC dataset that were measured using these cost functions show that our proposed **NeuPD** framework has outperformed existing approaches with an RMSE of 0.490 and R^2^ of 0.929.

## 1. Introduction

The field of precision medicine in cancer has attracted serious attention from researchers at different stages. The identification of underlying genomic features that lead to mutations has become an effective research area. The identification of specific mutations, methylation, copy number variants (CNVs), and gene expression are considered important contributors in cancer studies. The specific changes in particular genes may result in specific resistance to generally available drugs. On the other hand, as complexities develop at the individual level, they demand specific interventions for the particular target in the area of precision medicine. The identification of specific genes and the association of biomarkers with particular diseases require evidence-based studies to report the drug response accordingly. In the presence of hundreds of drugs, particularly for cancer, it is important to evaluate the specific response against the specific biomarkers and expression level of particular tumors. Moreover, to identify the most appropriate drugs, the method is being applied at the individual level for better results. This paper presents drug response prediction, which is a crucial topic in the field of cancer precision medicine. Precision medicine refers to the treatment that is best suited to the individual based on their multi-omics characteristics [1]. A drug might have a positive or negative effect on a patient’s body due to differences in individual biological characteristics. As conducting drug trials on each individual for large-scale research is expensive, time consuming, and challenging [2], we have worked on cell lines of tumor samples which are taken from the GDSC [3] and CCLE [4] datasets. These datasets contain multi-omics data such as gene expression, mutation, methylation, copy number variants, and so on for different cancer cell lines.

The deep learning-based techniques in the drug sensitivity prediction problem have been in use for a couple of years. Ref. [5] performed a systematic review of the literature covering 105 research papers that focused on using machine learning and deep learning techniques to predict the response of anti-cancer drugs. The review described multiple methods designed to enhance and categorize the response of cancer types to drug treatment. Due to the remarkable results of the deep learning models, we have also adopted the same technique. Our novelty in the proposed work is choosing the top 10 drugs’ chemical features with gene expression features to train our proposed model. These drugs have top modeling performance according to previous studies. This makes our proposed model simple and more robust in the drug sensitivity prediction problem.

Recently, the DREAM challenge has been opened, which attracts researchers and data scientists to use these datasets to solve the drug response prediction problem [6]. Many predictive models have been proposed to contribute to this problem, such as [7,8,9,10,11,12,13,14,15,16,17,18]. Some of the work includes anti-cancer drug repositioning [19,20,21]. Most of these works focus on the prediction of drug sensitivity to a panel of cancer cell lines using gene expression data. The drug sensitivity is measured in terms of the half-maximal inhibitory concentrations $IC_{50}$ value in (μM) units. In addition, the lower $IC_{50}$ value means that the drug is more sensitive or more effective against the disease, whereas a higher $IC_{50}$ value means drug resistance or that it is not effective.

Many predictive models have been used to solve the drug sensitivity prediction problem. All of these computation models use multiple omics datasets to predict drug sensitivity. Recently, ref. [22] proposed a set of guidelines for the proper use of machine learning models with gene expression data. Moreover, the emergence of these computational methods has had a significant influence on the identification of new applications for existing drugs [23]. Furthermore, these computational approaches have greatly facilitated a more systematic and rational approach to drug development processes, resulting in reduced timeframes for bringing drugs to market [24]. In summary, through the utilization of computational models and the integration of various data sources, these methodologies facilitate expedited and more effective drug screening, personalized treatment decision making, drug repurposing, prediction of drug toxicity, and identification of drug resistance. This advancement holds immense potential for advancing the efficiency and efficacy of healthcare interventions.

The authors in [25] proposed a DNN-PNN fresh parallel DL method to predict drug sensitivity by combining the strengths of two deep learning models. The first uses a DNN with a continuous gene expression profile as input, while the second uses a deep neural network based on a product and its discrete drug fingerprint traits. With higher prediction accuracy, faster convergence, and greater stability, the DNN-PNN outperforms both conventional and cutting-edge machine learning models, according to extensive experiments.

In [26], the authors used random forest to predict drug activity for cell lines based on chemical and genomic information. In contrast, Chiu et al. [27] recommended a DeepDR model that predicted drug response based on the cancer cell lines. This model comprises deep neural networks and a dataset taken from “TCGA: The Cancer Genome Atlas”. The model predicted values for 265 given drugs and was tested on 622 cell lines. The mean square error was 1.96 on the testing sample set. Another computational approach named ProGENI [28] detects genes that exhibit the strongest correlation with variations in drug response among diverse individuals. This method relies on gene expression data and distinguishes itself from existing approaches by additionally leveraging prior knowledge of protein–protein and genetic interactions. Random walk techniques are employed within ProGENI to incorporate this prior knowledge effectively. Another study was conducted using 678 drugs on only three cell lines. On both pathway and genetic levels, DNN performs better than the other model of SVM [29]. The limitation was that they used only three cell lines.

Deep learning is widely used to interpret the concepts based on deep layers to extract the representations of the data: see Refs. [30,31,32,33]. A deep learning model named DeepPredictor is proposed in [34] that predicts drug sensitivity using CCLE data. The results claim that the DeepPredictor performed better, and the coefficient of determination gave a value ranging from 0.68 to 0.75.

Another model named DeepDSC for drug sensitivity in cancer was proposed based on deep learning [35]. Their study obtained a dataset from GDSC and CCLE. The recommended model consisted of two stages. In the first stage, an autoencoder is used to extract the cell line features from gene expression data. In the second and final stage, the features will be presented as input to the deep neural network for sensitivity prediction. The validation was performed like [34] on the given datasets. The performance was measured by the coefficient of determination and the root mean square error (RMSE). The value of the coefficient of determination shows that it performed better than previous ones with R^2^ 0.78 for the GDSC and CCLE datasets; as we discussed in [34], the maximum value achieved was 0.75, indicating that DeepDSC might help cancer therapy and predict drug responses in the future.

An approach was taken for the purpose of classifying drug responses using regression. The proposed model AutoBorutaRF [36] was built to predict drug response. For the feature selection, the Boruta algorithm was used to improve the prediction. Dr.VAE [37] was a generative model that concurrently trains drug-induced changes on transcriptomic data and on drug response. As the term implies, the technique employs a VAE to generate latent representations of pre-treatment gene expression and match them with post-treatment gene expression in terms of prediction. In order to forecast drug reactions, both latent models are then put into a logistic regression classifier. Dr. VAE outperformed many traditional classification methods in terms of cross-validated AUROC ratings. As compared to our proposed work, Dr. VAE used drug sensitivity data as a classification problem, i.e., responders or non-responders, whereas we treated drug response as a regression problem because the discretizing sensitivity score loses some information.

MOLI [38] is another deep learning model that encodes mutation, copy number variants (CNVs), and gene expression data separately and concatenates them to represent cancer cell lines for drug classification. MOLI was trained on the GDSC dataset and tested on patient-derived xenografts. They have achieved up to a 0.75 AUC score on TCGA Cisplatin data with gene expression data. Whereas in our proposed method, we have reached a 0.490 IC_50_ value on the GDSC dataset for the top ten selected drugs, which is quite higher than many recent models.

A bit of a different approach was described in [39], which follows the deep learning concept along with a drug synergy named DeepSynergy. DeepSynergy is the mapping of input vectors to a single output that is known as the synergy score and predicts the scores of drugs for cancer cell lines. This proposed new model was compared with support vector machines, random forests, and boosting machines, and it proved to show better results with an increase of 7.2%.

For predicting drug response, a hybrid graph convolutional network model called DeepCDR was created [40]. This proposed method outperformed different techniques of classification and regression, such as ridge regression and random forest. The DeepDCDR achieved a 1.058 RMSE. A study of cultured cell line sensitivity and drug profiles was conducted to predict the inhibition patterns of the drugs along with the cell line profiles [10].

Although these methods can work reasonably well on CCLE and GDSC datasets with full genomic features set along with a maximum number of drugs, the downside of these methods is that they were limited to a casual integration of genomic and drug features. This drawback can lead to increased computational complexity and potentially make it difficult for readers to discern the key contributing factors in drug sensitivity prediction. Hence, there is a pressing need for a solution that enables a more comprehensive understanding of the integration between targeted drugs and genomic features while also facilitating the evaluation of biological significance.

To address this issue and take inspiration from the swift advancement of deep learning technology, this paper introduces a novel deep learning framework, **NeuPD**, designed for the prediction of drug sensitivity on cancer cell line data taken from GDSC and CCLE. Our approach involves the fusion of genomic profiles of cell lines and chemical profiles of compounds, forming a comprehensive architecture for predicting drug sensitivity.

In this research work, the data are extracted from GDSC, which is then preprocessed by applying a Mix–Max Scaler for normalization and then Pearson’s correlation for dimensionality reduction. Standard machine learning approaches, which are elastic net, XGBoost, and neural networks, are applied to the preprocessed data and then compared, giving rise to the proposed model, which is a deep neural network named **NeuPD** to assist medical practitioners and researchers in cancer therapeutics by working on essential cancer genes and the top 10 anti-cancer drugs. Different evaluation measures are used to verify the results.

To be precise, the novelty of our proposed model can be summarized in the following directions:Our contribution to modeling the drug sensitivity prediction problem is the selection of the top 10 drugs of interest. These drugs have been deduced from previous studies based on achieving top modeling performance. We took these drugs’ chemical features to integrate them with genomic features and trained our proposed deep neural network. The remarkable performance of our model shows that deep learning can enhance the work on drug response prediction. Our proposed approach has achieved outstanding results when compared with previous models.Our model has been applied to GDSC and CCLE datasets and was tested using RMSE, MAE, and R^2^ scores. By comparing the results, the lowest RMSE score shows the achievable part of our proposed model.

The structure of the rest of the paper is as follows: Section 2 presents the Material and Methods, Section 3 discusses the experimental environment, and Section 4 presents the results and discussion. Finally, the conclusion and future work are presented in Section 5.

## 2. Materials and Methods

### 2.1. Materials

To predict the drug sensitivity on unseen cancer cell lines, a deep learning neural network named NeuPD is proposed. As an input, gene expression data for cell lines, drug response data, and drug compound fingerprints were integrated into the model.

The drug response data were collected from the public repository of Genomics of Drug Sensitivity in Cancer (GDSC) (https://www.cancerrxgene.org/, accessed on 5 March 2023) and Cancer Cell Line Encyclopedia (CCLE), while data related to genes are collected from COSMIC Cancer Gene Census (https://cancer.sanger.ac.uk/cosmic/, accessed on 5 March 2023, [41]).

#### 2.1.1. GDSC

This is the largest publicly available repository for drug response in cell lines. The baseline dataset contains 17,737 genes from 1018 human cell lines, and the drug response data contain IC_50_ measurements of 198 anti-cancer drugs across more than 500 human cell lines for each of them, respectively. The IC_50_ was converted to Log_e_-IC_50_, and smaller values of Log_e_-IC_50_ mean a better response: that is, more cancer cells are dying. The compound structures for 10 drugs in 2D structures were collected from PubChem (https://pubchem.ncbi.nlm.nih.gov/, accessed on 5 March 2023, [42]) in Structure-Data File (SDF) which contains a compound structure-data file format that can link data with one or more chemical structures. It is commonly referred to as SDF, .sdf, or SD file. The structures are illustrated in Figure 1.

The collected structures were processed for hashed count Morgan fingerprints of 256 bits, making the obtained drug feature vectors length of 256. The final matrix consists of drug responses, gene expressions, and drug features which are then further processed for the training of the model. The list of all selected drugs along with their PubChem Compound ID number is given in Table 1.

#### 2.1.2. CCLE

This contains 16,383 gene identifiers of 1037 cell lines that were downloaded from the CCLE website. The drug responses were given in values of IC_50_. The 2D compound structures for 24 drugs were collected in the same way as GDSC, making the feature vector of size 256. The structures are illustrated in Figure 2.

The list of all selected drugs along with their PubChem Compound ID number is given in Table 2.

### 2.2. Datasets Preprocessing

Before the model training, we performed normalization to scale the numerical data without distorting its actual shape. For this step, we have used the Min–Max Scaling method using Equation (Equation 1) in which the data are scaled to a range of 0–1. Each value at the features is processed in a way that the minimum value will be transformed into the value of 0, and thus, the maximum value will scale to the value of 1.
(1)$$x_{scaledvalue}=\frac{x-min(x)}{max(x)-min(x)}$$

Equation (Equation 1) is the mathematical formula for the Min–Max Scaler where *x* is a single feature of the data.

After normalization, we have reduced the size, i.e., 10 drugs instead of 198 drugs from the GDSC dataset. We selected these drugs based on the top-ranked cancer drugs with the process defined in [43]. The selection of these drugs followed two criteria: Firstly, they were identified as the most suitable candidates for modeling based on all the feature selection methods employed. Secondly, they exhibited a noteworthy superiority in modeling when compared to the other type of feature selection methods, either genome-wide or biologically driven. Among the selected drugs, five demonstrated superior modeling performance when utilizing genome-wide features, whereas the remaining five exhibited better modeling outcomes with biologically driven features. This gives us the best top 10 modeled drugs. From the GDSC dataset, 851 cell lines against 10 selected drugs as response data have been extracted. From the CCLE dataset, we took 24 drug responses against 491 cell lines.

The total samples with all available drugs X cell−lines in the final data matrices are given in Table 3.

### 2.3. Cell Line Features Selection

For both CCLE and GDSC datasets, cell line features are expressed as a vector of gene expression values which are quite larger than the drug Morgan feature vector of 256 bits. So, to avoid the dominance of gene expression features over the drug features during model training, we have reduced the gene expression features by using dimensionality reduction. To achieve this objective, we used the Pearson correlation coefficient (PCC) method to obtain the optimized gene expression features up to 500 feature vectors. To find the correlation between all pairs of variables, a Pearson correlation matrix was calculated. This will give us a table of correlation coefficients, one for each pair of variables. Its range lies between −1 and 1, where −1 means a negative correlation, 0 means no correlation, and +1 means a positive correlation. PCC measures the linear relationship between two variables. The focus will be on variables having a high correlation coefficient. Equation (Equation 2) is the mathematical formula for calculating the Pearson correlation coefficient.
(2)$$P=\frac{\sum_{n=1}^{N}\left(x_{i}-x^{\prime }\right)\left(y_{i}-y^{\prime }\right)}{\sqrt{\sum_{n=1}^{N}{\left(x_{i}-x^{\prime }\right)}^{2}}\sqrt{\sum_{n=1}^{N}{\left(y_{i}-y^{\prime }\right)}^{2}}}$$

Let us assume that there are N samples, i.e., cancer cell lines in dataset S, where *P* is the correlation coefficient, *x_i_* here represents the gene expression values as *x*-variables in the sample, *x*’ is the mean of the gene expression values of the *x*-variables, *y_i_* represents the drug sensitivity (IC_50_) values of the *y*-variables to the panel of cancer cell lines samples, and *y*’ is the mean of the IC_50_ values of the *y*-variables. If *P* > 0, then it implies a positive relation, *P* < 0 implies that it is a negative correlation, while *P* = 0 represents that there is no correlation between the two variables.

### 2.4. NeuPD

With the increase of more neural units and layers, a feedforward deep neural network (DNN) would be able to produce a more universal approximator [44]. In many scientific areas, the powerful modeling technique has been extensively used [20,21,45,46]. These neural network-based modeling capabilities have been applied in drug sensitivity prediction problems for a couple of years. In spite of high-dimensional pharmacogenomics profile data, DNN has achieved remarkable results both in regression and classification tasks. In the regression task, DNN is used to predict drug sensitivity value in terms of IC_50_ value. While in the classification task, drug sensitivity values are discretized. However, ref. [14] has pointed out that the drug sensitivity prediction as a classification task loses more information than that of the regression task. Moreover, representing drug sensitivity data as a regression problem has also outperformed the classification problem. Based on these facts, we have used IC_50_ values as regression values to train our proposed deep learning model.

Figure 3 gives the overall workflow diagram of our proposed model, which is a NeuPD. The work is undertaken on two different datasets, i.e., GDSC and CCLE. These datasets contain different numbers of cell lines with gene expression values as cell line feature vectors. There were around 20,000 genes in both datasets, from which we further extracted 1610 essential oncogenes expressions based on CRISPR experiment [47]. The gene expressions were first normalized using the method defined in Equation (Equation 1), and then, the Pearson correlation method was applied to further reduce the cell lines features up to 500 features vector. The reason for this reduction is to make gene expression features close to the drug’s Morgan fingerprint features in terms of vector length. Moreover, to build final input matrices, drug features with known drug responses to a panel of cancer cell lines were integrated.

### 2.5. Experimental Environment

All methods are implemented using Python3 Scikit-learn, Keras, and TensorFlow libraries. For the process of forming hash values of Morgan fingerprints, an environment of RDKit is built for importing the required modules. The code is created in Python 3.9 on a 4.9 GHz Intel Core i7 CPU with 16 GB RAM equipped with high-performance NVIDIA GeForce MX130 GPU, using Google Colab and a Jupyter Notebook.

The architecture is then implemented using Keras embedded with four layers excluding the input and output layer, each having a weight less than the previous layer. The input dimension assigned in an input layer is 756, which comprises 500 dimensions for the reduced gene expressions data and 256 for the Morgan fingerprints. The first layer contains 1000 neural units, the second layer contains 800 neural units, the third layer contains 500 neural units, and the fourth contains 100 neural units. The activation function used for all hidden layers is ReLU (Rectified Linear Activation Function).

In the output layer, a weight of 1 neural unit is applied with no activation function, so the output range will not be restricted to a specific range. The loss function applied is of root mean square error to measure the magnitude of error in the applied model.

To overcome the issue of overfitting, early stopping is used with patience of 30, and a dropout rate of 0.1 is set after each layer. By the term dropout, it means to randomly select some neurons to omit during training.

#### Evaluation Measures

There are many different metrics used for measuring the performance of the model and measuring the magnitude of the error. The functions which are used to measure the error are known as loss functions. Some of the metrics used in this thesis work are RMSE, MAE, coefficient of determination (R^2^), and MSE.

RMSE Among all these metrics, RMSE is the loss function that is used to measure the error of the model as it is more suitable and accurate to measure the drug response model. RMSE is calculated by taking the difference between the predicted and target variable and then taking the root of the value, giving only positive values. The formula of RMSE is given in (Equation 3) where *y_i_* is the target variable that is drug response and *y_i_*^’^ is the predicted variable that is the features and *N* is the total size of the test data. (*y_i_*−*y_i_*^’^) is the target and predicted drug sensitivity data.
(3)$$\sqrt{\frac{\sum {\left(y_{i}-y_{i}^{\prime }\right)}^{2}}{N}}$$MSE: It is the measure of the average squared difference between the predicted values and the actual value.MAE: It is the mean absolute error that measures the magnitude of errors between the predicted and actual value of the dataset.$R^{2}$: It is the coefficient of determination and is used to measure the predictive performance in our model which is NeuPD. It explains how much divergence of one variable can occur by its relationship to the other variable.

To be specific with the training details, the Keras framework is used based on the backend of TensorFlow and validated with 10-fold cross-validation by taking one fold for validation and remaining folds for training that were repeated ten times, as illustrated in Figure 4.

We applied 10-fold cross-validation. The ReLU activation function is used in the full connection layer for the network structure in the feature extraction stage. The optimizer used is Adamax.

### 2.6. Summarized Model

A flowchart of the **NeuPD** is illustrated in Figure 3.

The process starts by gathering the data from different public repositories of GDSC and CCLE. The data are then normalized using Min–Max Scaler. For reducing the dimension, Pearson correlation is applied to reduce it to a dimension of 500. The datasets are preprocessed by extracting essential genes, the top 10 best-modeled drugs, and extracting the 2D structure of drugs from the PubChem repository. The 2D structures are converted into Morgan fingerprints to identify the drug features. The final matrix consists of chemical features that are the Morgan fingerprints and genomic features that are the gene expression. The drug features are of 256 bits, and the gene expression dimension as mentioned above is 500, making the final matrix of size 756. The final matrix consists of gene expressions and drug features versus the target variable, which is the drug response. The model is fitted by tuning different parameters, which are already defined. The fitted model is trained in the proposed NeuPD, and predictions are made by adding new data and evaluating the results using different metrics or loss functions of RMSE, MSE, MAE, and R^2^.

## 3. Results

To achieve less biased results in performance analysis, we used 10-fold cross-validation in performance analysis of the NeuPD trained on the datasets of GDSC and CCLE. An average is taken for all loss functions to achieve the final results accordingly. The mean of the values is taken across the genes, novel cell lines, and drug responses. Different ML-based techniques are used to validate and compare the results obtained on GDSC and CCLE datasets.

### 3.1. Baseline Methods

Apart from the proposed method, two competing machine learning techniques are also used to compare the results, which are outlined below.


**Elastic Net**
It is a regularized linear regression model having L1 and L2 penalty functions. It is a further extension to the regular linear regression by adding regularization penalties during the training phase to the loss function. It has two hyperparameters: *alpha* and *lambda*. *Alpha* can be set by declaring *l1_ratio* and *lambda* by *alpha*. A default value of 0.5 is assigned for *“l1_ratio”* and for *“alpha”*, a full weighting is used of 1.0, as there were no major changes in the results when different values were tried. Previous studies used the elastic net as a technique to predict drug response [48,49]. This elastic net model was implemented using the sklearn library applied to the processed dataset by using a repeated 10-fold cross-validation five times. To evaluate the results, loss functions were used of RMSE, MSE, MAE, and R^2^.
**XGBoost**
It is a gradient boosting library that is implemented under the framework of gradient boosting. It is beneficial as it provides a parallel way to solve data-related issues in a fast and precise way. This model supports Scikit-learn advanced features along with regularization. In a SciPy environment, it can be easily installed using pip. The command to install xgboost is: *sudo pip install xgboost*.XGBoost gives a wrapper class, which means that the models can be treated as classifiers or regressors in the Scikit framework. The models for the classifier and regressor are “XGBClassifier” and “XGBRegressor”. To make predictions, a **model.predict()** function is used, which is a Scikit-Learn built-in function. After fitting the model, evaluation is performed using different metrics.

### 3.2. Comparison on CCLE Dataset

As already discussed, the CCLE dataset consists of 16,383 gene expressions for 1037 cell lines and 24 drugs. The results achieved from cross-validation were compared with the state-of-the-art study [50] in which the CCLE dataset was used, giving an RMSE of 5.378 for the elastic net, while the RMSE achieved in our research is 3.202 for elastic net and 2.074 for XGBoost. NeuPD showed a much lower value of RMSE than the elastic net and XGBoost, which is 1.784. The methods comparison based on root mean square error (RMSE), mean absolute error (MAE), mean square error (MSE), and coefficient of determination (R^2^) is given in Table 4 below.

### 3.3. Comparison on GDSC Dataset

As already described above, these data contain 983 cell lines related to 1610 genes for 10 drugs. The final matrix thus is made up of drug responses, gene expressions, and Morgan fingerprint for 10 drugs that are 256 bits.

The results achieved outperformed the results by having an RMSE of 0.490, while the coefficient of determination R^2^ is 0.929. The closer the R^2^ is to 1, the better it fits the value. The RMSE achieved is 1.784 for CCLE. The lesser the value of RMSE, the better the model is because it means that the error value is less, and in precision medicine, we need an approach that gives less error as compared to other approaches.

The methods comparison based on RMSE, MAE, coefficient of determination (R^2^), and MSE is given in Table 5.

By comparing the results with two other approaches performed—elastic net and XGBoost—it can be seen that the elastic net again performs worse than XGBoost and NeuPD by having the RMSE value of 2.419 and R^2^ of 0.532, which means it did not fit the model well. Meanwhile, XGBoost performed much better than the elastic net, having an RMSE of 1.337 but for the R^2^, not much improvement was seen, as the value differs by only 0.1. The proposed model, again, outperformed the other baseline methods used in this research.

The results of both datasets are given in Figure 5.

### 3.4. Comparisons to Previous Studies

The RMSE achieved by applying NeuPD is 0.490 and the R^2^ achieved is 0.929. By looking at previous studies, the comparison can be made for the results achieved with other existing relevant models. All these existing studies used 10-fold cross-validation to calculate their RMSE scores. The RMSE achieved was 0.52 ± 0.01 and R^2^ was 0.78 ± 0.01 by using neural networks with 10-fold cross-validation [35]. A model named DeepCDR [40] achieved an RMSE of 1.058 ± 0.006. In a matrix factorization approach [51], an RMSE of 1.73 was achieved for the GDSC dataset. A study using the GDSC dataset along with Morgan fingerprints achieved a lower RMSE of 0.75 ± 0.01 and R^2^ of 0.74 ± 0.01 [10]. The results are given in Table 6 below.

The performance of the proposed NN-based NeuPD model was compared to the elastic net (EN) regression and XGBoost. By looking at the results, NeuPD achieved the best results compared with other baseline models and different studies. To lessen the chance of overfitting, early stopping is also used in which the training phase was stopped if the validation loss did not decrease after 500 epochs. The performance of each algorithm was evaluated by using the root mean square error (RMSE). The best value of RMSE is 0.490, which was achieved for the GDSC dataset over 10 drugs. For the CCLE dataset, the least value achieved for RMSE is 1.784. So, in these datasets, the proposed model performed better for GDSC than CCLE.

## 4. Discussion

The abundance of cancerous control and disease datasets at the genomic and proteomics levels allow us to better understand the drug and disease appropriateness. The genomic data including the gene expression level have been found to be an important biomarker toward the state and nature of cancer disease. The medicine prescription has entered the era of precision medicine that aimed to prescribe the most appropriate drug for the target disease sample based on the association with respective genomic biomarkers.

The availability of cancer cell lines, in particular, allowed AI experts to predict associations among hundreds of available drugs and associated genes through ML and DL tools. Several DL models have been proposed to identify the right drug for the GDSC and CCLE datasets. Drugs’ chemical compositions are being used to represent drug features.

There are numerous recent computational methods such as [7,8,9,35] that potentially contributed to the drug sensitivity prediction problem. Despite the effectiveness of these methods on CCLE and GDSC datasets using complete genomic features and a wide range of drugs, they have a limitation when it comes to integrating genomic and drug features causally. This limitation results in increased computational complexity and can make it challenging for the scientific community to identify the key factors influencing drug sensitivity prediction.

In this research work, we proposed a deep learning model named **NeuPD** that predicts the antineoplastic drug sensitivity of cancer cell lines. Our proposed model was trained on GDSC and CCLE datasets downloaded from the public repository. The data were normalized by applying Mix–Max Scaler, and then Pearson’s correlation was applied for dimensionality reduction. For the drug features, the 2D structures were downloaded for each drug from PubChem for both the GDSC and CCLE drugs. The structures were then converted into hash values using Morgan fingerprints. For the training of the model, gene expressions are selected as cell line genomic features, and Morgan fingerprints are selected as chemical features of compounds. Both cell line features and drug features were combined as the inputs for the NeuPD.

The results have shown that **NeuPD** outperformed previous state-of-the-art models and techniques by having the lowest value of RMSE which is 0.490 and the highest value of the coefficient of determination R^2^, which is 0.929. Moreover, the performance of the proposed model was evaluated using 10-fold cross-validation.

The present research work has also a few limitations, i.e., the size of the data and the applied DL tools employed during the course of the study. The applicability of our research findings is based on the available current state-of-the-art published results and the associated data. The biological scientific community is publishing tremendous results on an (almost) weekly basis, which demands the inclusion of the newly published results in a continuously updated mode. This inclusive nature demands extensive evaluation matrices of our results in consideration of the current state of the art. Secondly, high dimensionality is a challenge to the applicability of our research at a clinical level. In future, this work can be extended by integrating additional molecular features such as mutations, copy number variations, and methylation along with the gene expression data. Furthermore, incorporating explainability and interpretability into the framework would enable explicit extraction of biological significance from the obtained results.

## 5. Conclusions

In the recent past, the identification of appropriate drugs for particular genomic markers was found to be important in diagnostics and the prescription of drugs accordingly. Several machine learning and deep learning methods are being used to predict a significantly reduced subset of hundreds of available drugs. This study performed experiments on cancerous datasets extracted from different tissues and identified the most appropriate drugs for particular gene expression levels. The implemented and tested **NeuPD** method outperformed the existing state of the art and was helpful in predicting associations among the drugs for respective biomarkers across the different types of cancer disease. The study is aimed to develop results that have the potential to be employed by clinical practitioners in the future. 

## Figures and Tables

**Figure 1 diagnostics-13-02043-f001:**
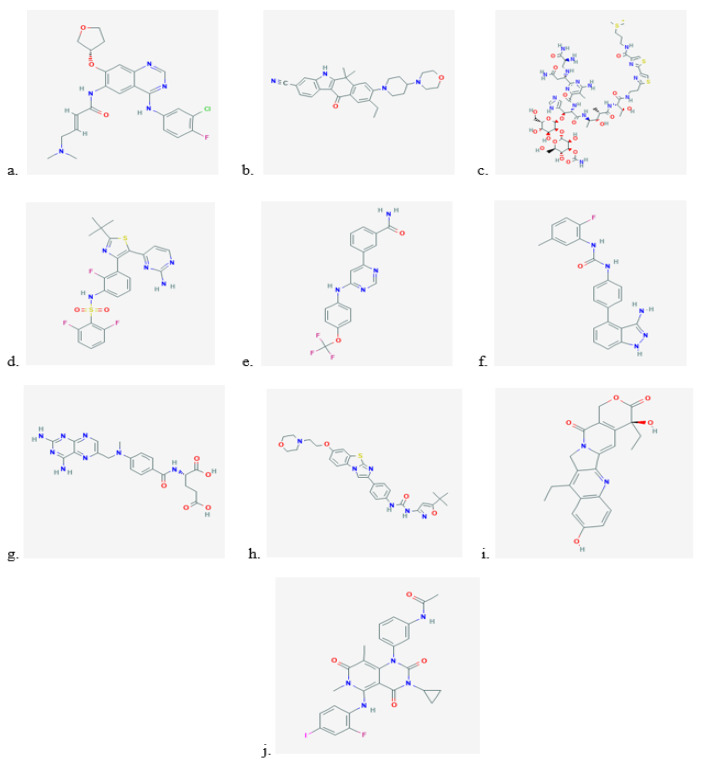
GDSC drugs 2D structures (**a**) Afatinib, (**b**) Alectinib, (**c**) Bleomycin, (**d**) Dabrafenib, (**e**) GNF-2, (**f**) Linifanib, (**g**) Methotrexate, (**h**) Quizartinib, (**i**) SN-38, and (**j**) Trametinib.

**Figure 2 diagnostics-13-02043-f002:**
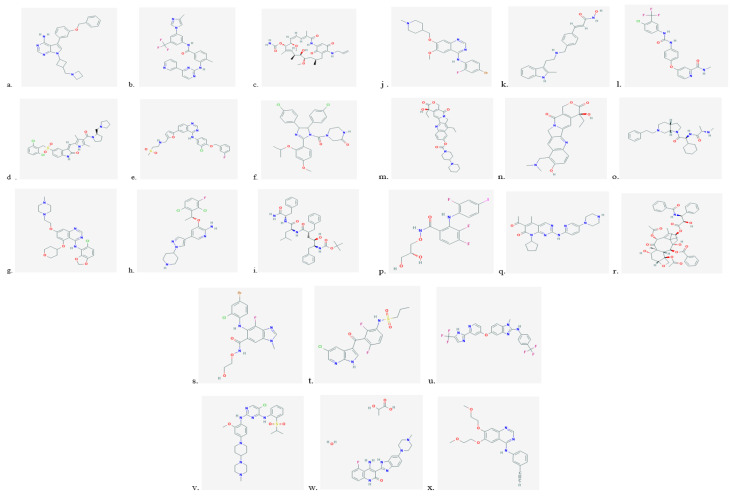
CCLE drugs 2D structures: (**a**) AEW541, (**b**) Nilotinib, (**c**) 17-AAG, (**d**) PHA-665752, (**e**) Lapatinib, (**f**) Nutlin-3, (**g**) AZD0530, (**h**) PF2341066, (**i**) L-685458, (**j**) ZD-6474, (**k**) Panobinostat, (**l**) Sorafenib, (**m**) Irinotecan, (**n**) Topotecan, (**o**) LBW242, (**p**) PD-0325901, (**q**) PD-0332991, (**r**) Paclitaxel, (**s**) AZD6244, (**t**) PLX4720, (**u**) RAF265, (**v**) TAE684, (**w**) TKI258, (**x**) Erlotinib.

**Figure 3 diagnostics-13-02043-f003:**
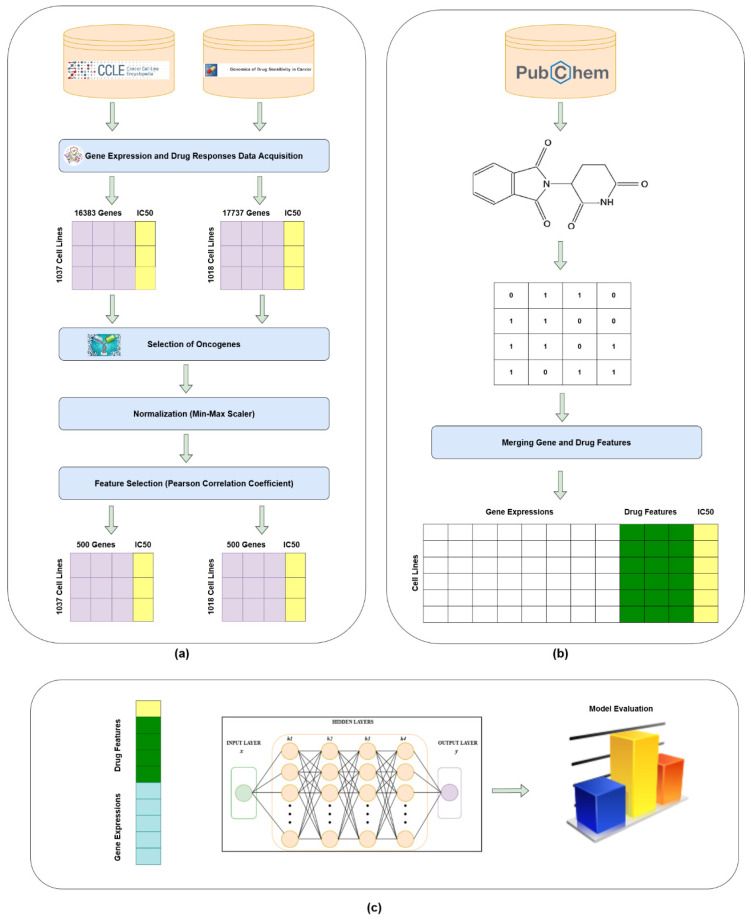
Diagram of the NeuPD model. (**a**) The process commences with data acquisition from the CCLE and GDSC datasets, which is followed by a selection of oncogenes’ expression data and then data normalization followed by PCC-based gene feature selection. (**b**) Extracting 2D drug structures from PubChem datasets, converting them into Morgan fingerprints as drug features and finally merging them with gene expression data to produce the final response data. (**c**) Both genomic and drug features are concatenated together as the input to train the neural network model. The model performance is evaluated in terms of RMSE and R^2^.

**Figure 4 diagnostics-13-02043-f004:**
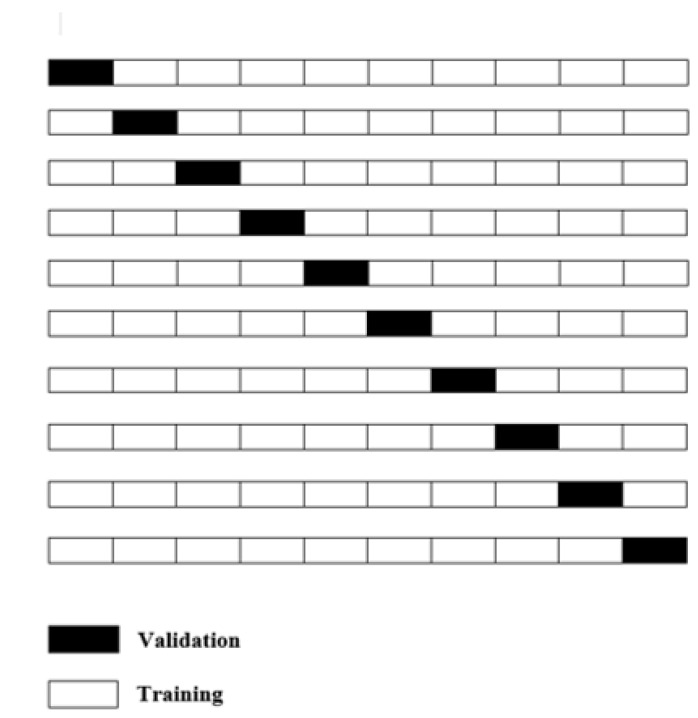
10-fold cross-validation.

**Figure 5 diagnostics-13-02043-f005:**
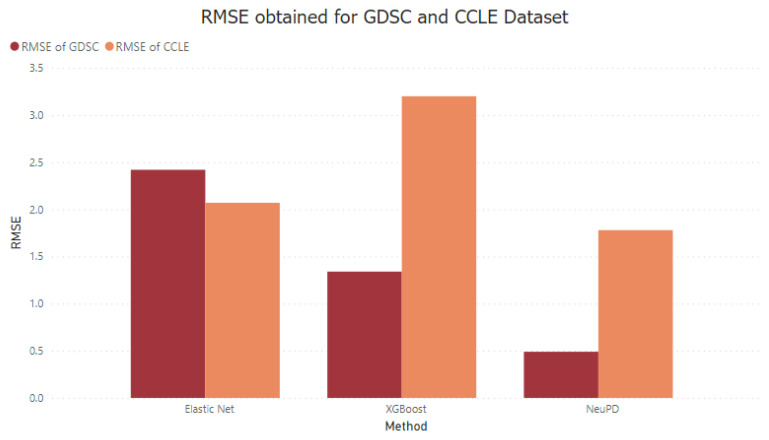
RMSE comparison for GDSC and CCLE dataset.

**Table 1 diagnostics-13-02043-t001:** GDSC drug names and PubChem ID.

Drug Name	PubChem CID
Afatinib	10184653
Alectinib	49806720
Bleomycin (50 uM)	5460769
Dabrafenib	44462760
GNF-2	5311510
Linifanib	11485656
Methotrexate	126941
Quizartinib	24889392
SN-38	104842
Trametinib	11707110

**Table 2 diagnostics-13-02043-t002:** CCLE drug names and PubChem ID.

Drug Name	PubChem ID	Drug Name	PubChem ID
AEW541	11476171	Irinotecan	60838
Nilotinib	644241	Topotecan	60700
17-AAG	6505803	LBW242	11503417
PHA-665752	10461815	PD-0325901	9826528
Lapatinib	208908	PD-0332991	5330286
Nutlin-3	216345	Paclitaxel	36314
AZD0530	10302451	AZD6244	10127622
PF2341066	11626560	PLX4720	24180719
L-685458	5479543	RAF265	11656518
ZD-6474	3081361	TAE684	16038120
Panobinostat	6918837	TKI258	135611162
Sorafenib	216239	Erlotinib	176870

**Table 3 diagnostics-13-02043-t003:** Preprocessed Data.

Dataset	Drugs	Cell Lines	Data Points
GDSC	10	851	8510
CCLE	24	491	11,784

**Table 4 diagnostics-13-02043-t004:** Results on CCLE Dataset.

Method	RMSE	MSE	MAE	$R^{2}$
Elastic Net	±3.202	±10.25	±2.972	0.281
XGBoost	±2.074	±4.308	±1.491	0.317
**NeuPD**	±**1.784**	±**3.192**	±**1.568**	**0.543**

**Table 5 diagnostics-13-02043-t005:** Results on GDSC Dataset.

Method	RMSE	MSE	MAE	$R^{2}$
Elastic Net	±2.419	±5.851	±2.020	0.532
XGBoost	±1.337	± 1.794	±0.953	0.609
**NeuPD**	±**0.490**	±**0.246**	±**0.392**	**0.929**

**Table 6 diagnostics-13-02043-t006:** Results on GDSC Dataset.

Dataset	Method	RMSE
**GDSC**	SRMF [51]	±1.73
	DeepCDR [40]	±1.058
	PGM [10]	±0.75
	DeepDSC [35]	±0.52
	**NeuPD**	±**0.490**
**CCLE**	Ridge Regression [50]	±6.576
	Random Forest [50]	±5.738
	Elastic Net [50]	±5.378
	Lasso [50]	±5.333
	**NeuPD**	±**1.784**

## Data Availability

The datasets used in this research are available at https://www.cancerrxgene.org, https://cancer.sanger.ac.uk/cosmic, https://pubchem.ncbi.nlm.nih.gov, https://drive.google.com/drive/folders/1RqPDQ5eKAEAG1i5hQpHFv2qBQ_qnSKJw?usp=sharing (accessed on 5 March 2023).

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
