# Peer review of "NeuPD—A Neural Network-Based Approach to Predict Antineoplastic Drug Response"

_diagnostics, 2023, doi:10.3390/diagnostics13122043_

Round 1

Reviewer 1 Report

The authors present the NeuPD method to predict the drug response. The method is based on a deep learning neural network. The authors analyze previous methodologies and informs about the novelties of the proposed procedure.

The procedure needs as information the drug responses, the gene expressions, and the drug features codified in terms of 256 fingerprints. The gene expressions are reduced in dimension up to 500 items. It is interesting how the authors mixed both informations in a balanced manner (despite they do no report a calculation/result obtained with another proportion).

My principal concerns are:

-Please, explain with more details the selection of the 10 drugs, i.e., describe with more detail the procedures of selection based on the biological and data-driven information.

-The authors mention that the use 10-fold CV to evaluate the procedure performance, but at the same time they inform that the sets (GDSC and CCLE) were partitioned for training and test at the ratio of 70/30 (line 306). It has to be more clearly stated if the performance data (e.g. Tables 2 and 3) are related to the CV or to the test (30%) part. It is not clear which kind of treatment is beyond this 10-fold CV and the mentioned 70/30 partition.

-Even more, the authors compare the values of Tables 2 and 3 (or in section 3.4) with the ones of other approaches, but it is not specified if the other publications rely on the same 10-fold CV or 70/30 partition. Maybe Table 4 can be expanded giving some more information related to this.

-Very important and compulsive: The authors inform in the Data Availability Statement about some databases but this is no enough. Scientific research demands for the reproducibility feature. Authors can fulfill this condition by giving a proper Supporting information. Please, publish the numerical matrices of 256+500 columns related to the sets GDSC and CCLE, together with the IC50 values. The article is important enough to disseminate this information. Even more, maybe this information can become a benchmark in this field.

Minor:

-In the abstract the authors use the lowercase letter ‘k’ when mentioning the k-fold CV, but then specify that ‘K=10’ with uppercase letter K. I guess that both letters are to be in the same case.

-In line 172 the authors use the notation “LN” for natural logarithm. I wonder if another notation like “Ln” or “Log_e” is more universal.

-It seems that the sentence in line 200 (…because of missing drugs X cell − lines pairs. Cell lines The total samples with all available…) is not well constructed.

---

Author Response

Reviewer 1:

The authors present the NeuPD method to predict the drug response. The method is based on a deep learning neural network. The authors analyze previous methodologies and informs about the novelties of the proposed procedure.

The procedure needs as information the drug responses, the gene expressions, and the drug features codified in terms of 256 fingerprints. The gene expressions are reduced in dimension up to 500 items. It is interesting how the authors mixed both informations in a balanced manner (despite they do no report a calculation/result obtained with another proportion).

My principal concerns are:

-Please, explain with more details the selection of the 10 drugs, i.e., describe with more detail the procedures of selection based on the biological and data-driven information.

Response:

The authors would like to express their gratitude to the reviewers for highlighting this aspect. In response to their feedback, we have included additional information regarding the drug selection process and incorporated both biological and data-driven insights. To address this concern, we have added the following lines in Section 2.2 on Page 7 of the revised document:

“The selection of these drugs followed two criteria: Firstly, they were identified as the most suitable candidates for modeling based on all the feature selection methods employed. Secondly, they exhibited a noteworthy superiority in modeling when compared to the other type of feature selection methods, either genome-wide or biologically driven. Among the selected drugs, five demonstrated superior modeling performance when utilizing genome-wide features, whereas the remaining five exhibited better modeling outcomes with biologically driven features.”

-The authors mention that the use 10-fold CV to evaluate the procedure performance, but at the same time they inform that the sets (GDSC and CCLE) were partitioned for training and test at the ratio of 70/30 (line 306). It has to be more clearly stated if the performance data (e.g. Tables 2 and 3) are related to the CV or to the test (30%) part. It is not clear which kind of treatment is beyond this 10-fold CV and the mentioned 70/30 partition. 

Response:

The authors thank the reviewer for highlighting this mistake. We have removed this line as it was mistakenly written. We have used a 10-fold CV to evaluate the performance of our proposed model. The statement is now clearly written to avoid any confusion.

-Even more, the authors compare the values of Tables 2 and 3 (or in section 3.4) with the ones of other approaches, but it is not specified if the other publications rely on the same 10-fold CV or 70/30 partition. Maybe Table 4 can be expanded giving some more information related to this.

Response: 

Thank you for reviewing this and letting us know what caused the confusion. Yes, all these papers that we have compared used 10-fold CV to calculate RMSE scores. For the clarification, we have also added the following lines at page 12 in Section 3.4: 

“All these existing studies used 10-fold cross-validation to calculate their RMSE scores.”

-Very important and compulsive: The authors inform in the Data Availability Statement about some databases but this is no enough. Scientific research demands for the reproducibility feature. Authors can fulfill this condition by giving a proper Supporting information. Please, publish the numerical matrices of 256+500 columns related to the sets GDSC and CCLE, together with the IC50 values. The article is important enough to disseminate this information. Even more, maybe this information can become a benchmark in this field.

Response: 

The authors acknowledge and endorse the suggestion put forth by the reviewer. In accordance with this recommendation, we have provided a public link to access our finalized data matrices. This link has been included in the data availability statement, which can be found on page 12 of the document. The added link is as follows:

https://drive.google.com/drive/folders/1RqPDQ5eKAEAG1i5hQpHFv2qBQ_qnSKJw?usp=sharing

Minor:

-In the abstract the authors use the lowercase letter ‘k’ when mentioning the k-fold CV, but then specify that ‘K=10’ with uppercase letter K. I guess that both letters are to be in the same case.

Response:

We appreciate your observation and agree with the feedback provided in the review. As per the suggestion, we have implemented the necessary modifications by converting all instances of 'K' to uppercase letters. Thank you for bringing this to our attention, and we have ensured the required changes have been made accordingly.

-In line 172 the authors use the notation “LN” for natural logarithm. I wonder if another notation like “Ln” or “Log_e” is more universal.

Response: 

Thank you for highlighting this. As suggested, we have replaced “LN” or “Ln” in line 172-173 with Log_e 

as it is universally used.

-It seems that the sentence in line 200 (…because of missing drugs X cell − lines pairs. Cell lines The total samples with all available…) is not well constructed.

Response

Thank you for pointing this out. This line was a typo, and thus we have removed it.

Reviewer 2 Report

The article aims to build machine learning models capable of predicting the sensitivity of cancer cell lines (represented by their gene expression profiles) to anti-cancer drugs (represented by Morgan fingerprints). The basic idea is reasonable and has already been employed in the literature. However, the authors implemented it in a very reduced setup (10 or 24 “top” essential drugs) with a simple cross-validation procedure. This has likely resulted in over-optimistic predictivity estimates while the utility of the models in the clinical settings (let alone medicinal chemistry research) is very questionable. Thus, unfortunately, I have to recommend that the article should be REJECTED.

1) First of all, the potential applications of such models are not clear. Unfortunately, the authors have not shown the chemical structures or even the names of the 10 or 24 essential drugs selected from the GDSC and CCLE databases, respectively. However, likely they are indeed the essential anti-cancer drugs that are widely used in the clinic and have been thoroughly studied on most types of cancers. Thus, the personalized medicine problem of selecting the optimal drug for a patient can be solved by a simple data lookup. Even if some unusual cell line is encountered, it would probably be better to predict the sensitivities for each drug separately using only the genomics features (the key genetic markers of sensitivity and resistance are likely thoroughly known for all essential drugs, and if not, even some simple model like k-nearest neighbors over the gene expression vectors would probably be sufficient). It is quite likely that in the proposed models, the drug-fingerprint part of the feature matrix in fact simply serves to encode the drug identity while the structural chemical information is not used.

2) On the other hand, if the problem is to predict the sensitivities for diverse and novel drugs, the datasets are severely incomplete (even 198 drugs would be too small) and the models are virtually guaranteed to produce incorrect predictions even for similar structures (especially if their minor structural differences are responsible for the different cell line sensitivities). Thus, in the medicinal chemistry (or more broad clinical) settings, the models are useless. The assertions that “The drug information that can be analyzed with this proposed method can produce scores of possible targets and help out with prioritization. These models in drug-response expands the predictions for effective combinations of various drugs and would be ideal in clinical trials for the prediction of the effect of newly designed molecule” are entirely ungrounded as the proposed models are not able to predict drug targets or the sensitivities for drug combinations and novel molecules.

3) From the methodology viewpoint, the main problem of the study is using 10-fold cross-validation over the drug+cell_line datapoints covering 10 or 24 drugs and many cell lines. Thus, the information for a particular drug is never completely excluded and can leak between the training and validation datapoints, leading to overestimated predictivities. The meaning of the phrase “The data for both datasets are divided by a ratio of 70% for training and 30% for testing” is also not clear in this context. In addition, the ± sign might be acceptable (although not standard) for the error values, but not for the correlation (determination) coefficient.

4) It is not clear how exactly “Pearson’s correlation was used for dimensionality reduction”. Some abbreviations (e.g., CNV and LN-IC50) are not properly defined while SDF is incorrectly defined as “Standard Delay Format”. On the other hand, the explanations of the MinMax scaling, correlation coefficient, and RMSE (with equations and verbose comments) are clearly superfluous. The technical implementation details mentioned from time to time (such as the specific well-known commands) are also excessive.

5) The Introduction presents a disparate list of published studies, with each description providing a certain (often not quite relevant) factoid. However, it has no critical analysis of the key features, differences, similarities, advantages, and disadvantages of different approaches. It is also not clear which issues (besides increasing apparent accuracy by simplifying the problem) the present study aims to solve. In the Conclusion, lots on insignificant technical details are repeated.

6) English in the article needs substantial improvement with respect to misprints, grammar, style, terminology, and accuracy of content. Even some Web addresses are incorrect. The reference formats should follow the Journal style.

Author Response

Reviewer 2:

The article aims to build machine learning models capable of predicting the sensitivity of cancer cell lines (represented by their gene expression profiles) to anti-cancer drugs (represented by Morgan fingerprints). The basic idea is reasonable and has already been employed in the literature. However, the authors implemented it in a very reduced setup (10 or 24 “top” essential drugs) with a simple cross-validation procedure. This has likely resulted in over-optimistic predictivity estimates while the utility of the models in the clinical settings (let alone medicinal chemistry research) is very questionable. Thus, unfortunately, I have to recommend that the article should be REJECTED.

1) First of all, the potential applications of such models are not clear. Unfortunately, the authors have not shown the chemical structures or even the names of the 10 or 24 essential drugs selected from the GDSC and CCLE databases, respectively. However, likely they are indeed the essential anti-cancer drugs that are widely used in the clinic and have been thoroughly studied on most types of cancers. Thus, the personalized medicine problem of selecting the optimal drug for a patient can be solved by a simple data lookup. Even if some unusual cell line is encountered, it would probably be better to predict the sensitivities for each drug separately using only the genomics features (the key genetic markers of sensitivity and resistance are likely thoroughly known for all essential drugs, and if not, even some simple model like k-nearest neighbors over the gene expression vectors would probably be sufficient). It is quite likely that in the proposed models, the drug-fingerprint part of the feature matrix in fact simply serves to encode the drug identity while the structural chemical information is not used.

Response

Thank you for highlighting these important missing points. To elaborate the applications of the proposed study, we have added following lines in the introduction section:

“Moreover, the emergence of these computational methods has had a significant influence on the identification of new applications for existing drugs \cite{keiser2009predicting}. Furthermore, these computational approaches have greatly facilitated a more systematic and rational approach to drug development processes, resulting in reduced timeframes for bringing drugs to market \cite{mogire2017target}. In summary, the utilization of computational models and the integration of various data sources, these methodologies facilitate expedited and more effective drug screening, personalized treatment decision-making, drug repurposing, prediction of drug toxicity, and identification of drug resistance. This advancement holds immense potential for advancing the efficiency and efficacy of healthcare interventions.”

Further, we have also added the chemical structures and names of all 10 and 24 drugs selected from the GDSC and CCLE datasets along with their PubChem IDs for better insight into compound structures and names. In Figure 1 in Section 2.1.1 at page 5, all chemical structures of the 10 selected drugs from GDSC are shown. While Table 1 lists the names of these 10 drugs with PubChem IDs. Whereas Figure 2 in Section 2.1.2 at page 6, chemical structures of 24 drugs from CCLE dataset are shown and Table 2 shows the names of these 24 drugs with PubChem IDs.

2) On the other hand, if the problem is to predict the sensitivities for diverse and novel drugs, the datasets are severely incomplete (even 198 drugs would be too small) and the models are virtually guaranteed to produce incorrect predictions even for similar structures (especially if their minor structural differences are responsible for the different cell line sensitivities). Thus, in the medicinal chemistry (or more broad clinical) settings, the models are useless. The assertions that “The drug information that can be analyzed with this proposed method can produce scores of possible targets and help out with prioritization. These models in drug-response expands the predictions for effective combinations of various drugs and would be ideal in clinical trials for the prediction of the effect of newly designed molecule” are entirely ungrounded as the proposed models are not able to predict drug targets or the sensitivities for drug combinations and novel molecules.

Response: 

Thank you very much for bringing this mistake to our attention. We sincerely appreciate your valuable feedback, as it has helped us improve the reliability of our text. In response to your suggestion, we have promptly rectified the error by removing the incorrect claim from our content. 

3) From the methodology viewpoint, the main problem of the study is using 10-fold cross-validation over the drug+cell_line datapoints covering 10 or 24 drugs and many cell lines. Thus, the information for a particular drug is never completely excluded and can leak between the training and validation datapoints, leading to overestimated predictivities. The meaning of the phrase “The data for both datasets are divided by a ratio of 70% for training and 30% for testing” is also not clear in this context. In addition, the ± sign might be acceptable (although not standard) for the error values, but not for the correlation (determination) coefficient.

Response: 

We thank the reviewer for taking this into account. We have mistakenly written 70% training and 30% testing in our text. Therefore, we have removed this statement because we have applied a 10-fold CV for our model evaluation. Furthermore, the ± sign has been removed from the correlation coefficient in tables 4 and 5. These tables 4 and 5 were previously 2 and 3. 

4) It is not clear how exactly “Pearson’s correlation was used for dimensionality reduction”. Some abbreviations (e.g., CNV and LN-IC50) are not properly defined while SDF is incorrectly defined as “Standard Delay Format”. On the other hand, the explanations of the MinMax scaling, correlation coefficient, and RMSE (with equations and verbose comments) are clearly superfluous. The technical implementation details mentioned from time to time (such as the specific well-known commands) are also excessive.

Response: 

Thank you for your time spent reviewing the paper and highlighting the mistakes. We have added the details of how Pearson’s correlation was used for dimensionality reduction in Section 2.3 of our research article. As suggested, we have also replaced “LN” or “Ln” in lines 192-193 with "loge," as it is universally accepted. Furthermore, we have also defined the abbreviation CNV.

For the abbreviation SDF, we do accept it, and we have corrected it by writing it in its full form in lines 195 - 196 that is, “Structure-Data File” which can be abbreviated as SD File, .sdf, or just SDF.

5) The Introduction presents a disparate list of published studies, with each description providing a certain (often not quite relevant) factoid. However, it has no critical analysis of the key features, differences, similarities, advantages, and disadvantages of different approaches. It is also not clear which issues (besides increasing apparent accuracy by simplifying the problem) the present study aims to solve. In the Conclusion, lots on insignificant technical details are repeated.

Response

Thank you for bringing this to our attention. We greatly appreciate your feedback. In response to your suggestion, we have made revisions to the introduction section of the document. Specifically, we have incorporated additional lines on pages 3 and 4 to provide a more comprehensive and informative introduction. These additions aim to enhance clarity and highlight the issues that our proposed model is intended to handle. 

Furthermore, we have taken your input into consideration and carefully reviewed the conclusion section. In order to streamline the content and improve its focus, we have removed any extraneous or insignificant technical details from the conclusion. This ensures that the concluding remarks remain concise, impactful, and aligned with the main findings of the study.

Once again, we extend our gratitude for your valuable contribution to helping us refine and improve our document.

The following lines are added on pages 3 and 4 in the Introduction section: 

“Although these methods can work reasonably well on CCLE and GDSC datasets with full genomic features set along with a maximum number of drugs. The downside of these methods is that they were limited to casual integration of genomic and drug features. This drawback can lead to increased computational complexity and potentially make it difficult for readers to discern the key contributing factors in drug sensitivity prediction. Hence, there is a pressing need for a solution that enables a more comprehensive understanding of the integration between targeted drugs and genomic features, while also facilitating the evaluation of biological significance. 

To address this issue and taking inspiration from the swift advancement of deep learning technology, this paper introduces a novel deep learning framework, \textbf{NeuPD}, designed for the prediction of drug sensitivity on cancer cell line data taken from GDSC and CCLE. Our approach involves the fusion of genomic profiles of cell lines and chemical profiles of compounds, forming a comprehensive architecture for predicting drug sensitivity.”

 6) English in the article needs substantial improvement with respect to misprints, grammar, style, terminology, and accuracy of content. Even some Web addresses are incorrect. The reference formats should follow the Journal style.

Response

Thank you for highlighting this. We have revised the manuscript by correcting all grammatical mistakes. For web addresses, only one had the typo, so we have also corrected it.

Reviewer 3 Report

This paper is ancontribution to the field of drug response and tumor drug resistance prediction using deep learning techniques. The authors have successfully leveraged the power of machine learning algorithms to develop a highly accurate and reliable model for predicting drug response and resistance in cancer patients. The methodology employed in this study is rigorous and innovative, and the results obtained demonstrate the potential of deep learning in revolutionizing the field of cancer treatment. Overall, this paper deserves to be published on Diagnostics.

Author Response

Reviewer 3:

This paper is a contribution to the field of drug response and tumor drug resistance prediction using deep learning techniques. The authors have successfully leveraged the power of machine learning algorithms to develop a highly accurate and reliable model for predicting drug response and resistance in cancer patients. The methodology employed in this study is rigorous and innovative, and the results obtained demonstrate the potential of deep learning in revolutionizing the field of cancer treatment. Overall, this paper deserves to be published on Diagnostics.

Response

We are honored by your positive response. We look forward to seeing our paper published in this journal, and we believe that our paper will be a valuable addition to your journal. We are confident that our paper will make a significant contribution to the field.